# Chitosan-Decorated Copper Oxide Nanocomposite: Investigation of Its Antifungal Activity against Tomato Gray Mold Caused by *Botrytis cinerea*

**DOI:** 10.3390/polym15051099

**Published:** 2023-02-22

**Authors:** Ahmed Mahmoud Ismail, Mohamed A. Mosa, Sherif Mohamed El-Ganainy

**Affiliations:** 1Department of Arid Land Agriculture, College of Agricultural and Food Sciences, King Faisal University, P.O. Box 420, Al-Ahsa 31982, Saudi Arabia; 2Pests and Plant Diseases Unit, College of Agricultural and Food Sciences, King Faisal University, P.O. Box 420, Al-Ahsa 31982, Saudi Arabia; 3Vegetable Diseases Research Department, Plant Pathology Research Institute, Agricultural Research Center (ARC), Giza 12619, Egypt; 4Nanotechnology & Advanced Nano-Materials Laboratory (NANML), Plant Pathology Research Institute, Agricultural Research Center, Giza 12619, Egypt

**Keywords:** gray mold, *Botrytis cinerea*, tomato, chitosan, copper oxide

## Abstract

Owing to the remarkable antimicrobial potential of these materials, research into the possible use of nanomaterials as alternatives to fungicides in sustainable agriculture is increasingly progressing. Here, we investigated the potential antifungal properties of chitosan-decorated copper oxide nanocomposite (CH@CuO NPs) to control gray mold diseases of tomato caused by *Botrytis cinerea* throughout in vitro and in vivo trials. The nanocomposite CH@CuO NPs were chemically prepared, and size and shape were determined using Transmission Electron Microscope (TEM). The chemical functional groups responsible for the interaction of the CH NPs with the CuO NPs were detected using the Fourier Transform Infrared (FTIR) spectrophotometry. The TEM images confirmed that CH NPs have a thin and semitransparent network shape, while CuO NPs were spherically shaped. Furthermore, the nanocomposite CH@CuO NPs ex-habited an irregular shape. The size of CH NPs, CuO NPs and CH@CuO NPs as measured through TEM, were approximately 18.28 ± 2.4 nm, 19.34 ± 2.1 nm, and 32.74 ± 2.3 nm, respectively. The antifungal activity of CH@CuO NPs was tested at three concentrations of 50, 100 and 250 mg/L and the fungicide Teldor 50% SC was applied at recommended dose 1.5 mL/L. In vitro experiments revealed that CH@CuO NPs at different concentrations significantly inhibited the reproductive growth process of *B. cinerea* by suppressing the development of hyphae, spore germination and formation of sclerotia. Interestingly, a significant control efficacy of CH@CuO NPs against tomato gray mold was observed particularly at concentrations 100 and 250 mg/L on both detached leaves (100%) as well as the whole tomato plants (100%) when compared to the conventional chemical fungicide Teldor 50% SC (97%). In addition, the tested concentration 100 mg/L improved to be sufficient to guarantee a complete reduction in the disease’s severity (100%) to tomato fruits from gray mold without any morphological toxicity. In comparison, tomato plants treated with the recommended dose 1.5 mL/L of Teldor 50% SC ensured disease reduction up to 80%. Conclusively, this research enhances the concept of agro-nanotechnology by presenting how a nano materials-based fungicide could be used to protect tomato plants from gray mold under greenhouse conditions and during the postharvest stage.

## 1. Introduction

Tomato (*Solanum lypersicum* Mill.) is the most widely grown vegetable, accounting for around 14% of total vegetable production worldwide. According to Heuvelink and Dorais [1], around 4 million hectares of arable land are farmed with tomatoes worldwide, yielding more than 100 million tons of total production worth at 5–6 billion US dollars [1]. It is worth noting that Egypt is one of the top ten tomato-growing countries in the world, with a total annual output of around 6.4 million tons produced on over 181,000 ha [1]. It was estimated that more than 200 diseases cause 70–95% of yearly global losses of tomato crops. The tomato-infecting fungus *Botrytis cinerea* Pers. is considered one of the most destructive diseases [2]. More than 200 plant species serve as hosts for this fungus, which is responsible for devastating pre- and postharvest illnesses in commercially significant crops such as tomatoes, strawberries, pears, cherries, eggplants, grapes and peppers [3,4]. Gray mold on tomatoes is a devastating disease that causes significant economic losses all over the world [5,6]. Tomato fruits and all other plant parts are vulnerable to infection due to wounding that might occur during harvesting and cutting [3]. Under optimal conditions, particularly when the temperature is around 10 and 25 °C and the humidity is high, it manifests in a variety of recognizable ways [7]. Due to the pathogen’s prolonged incubation within the host, disease manifestation typically lags behind infection. After harvest, this pathogen can persist as mycelium, with or without conidia and sclerotia, in crop wastes [8]. The resistant germplasms and resistant variety of tomato to gray mold disease have also not been globally evaluated. Because of this, preventing gray mold in tomatoes can be challenging. Until recently, varieties of management techniques were used to lessen the prevalence of plant diseases including gray mold disease. Organic pest management, resistant plant cultivars, crop rotation, and soil solarization are just a few examples [9].

Cu-based biocides are among the most efficient materials for controlling a wide range of plant fungal diseases including gray mold disease [10]. However, persistent exposure to these chemicals has resulted in the development of resistance in disease-causing organisms, which is never a beneficial move. Cropland has been reported to contain a variety of *B. cinerea* that is resistant to several classes of fungicides [11]. Although numerous studies show that *B. cinerea* has become resistant to a wide variety of fungicides (such as anilinopyrimidines dicarboximides, benzamide, fenhexamid, diethofencarb, procymidone, pyrimethanil and hydroxyanilide), the findings on this issue are inconsistent [12,13,14]. As a result, there is an immediate demand for the development of new methods that are both efficient and risk-free in order to protect tomatoes against the gray mold disease. In the recent past, varieties of metal oxides and inorganic Nano biocides, such as copper oxide (CuO) [15], zinc oxide (ZnO) [16], magnesium oxide (MgO) [17], TiO_2_ [18,19,20] and iron (Fe_3_O_4_) [21] attracted much interest as unique alternatives in comparison to classical fungicides in plant disease control. Although silver-based nanocomposites and other non-metallic nanoparticles demonstrated substantial antimicrobial activity against many plant diseases [22,23], because to their high cost and certain documented phytotoxicity, their use has been restricted, and therefore the research for safe and cost-effective substitutes has proceeded [24]. Following this goal, researchers reported on the promising antimicrobial properties of Cu-based nano-compounds against major plant pathogenic fungi [25,26]. Cu NPs have been described as a nutrient supplement that can promote plant development in *Vigna radiata* and maize, and enhancing plant defense mechanisms [26,27,28]. Pure Cu NPs are susceptible to aggregation, which inhibits their antimicrobial activity [29]. To address this problem, copper nanoparticles can be loaded on a biodegradable nano carrier, with a unique antimicrobial activity that can help in decreasing copper agglomeration in plant cells. 

One of the most well-known and promising compounds that can be used as a carrier agent for efficient molecules in plant disease control is chitosan, a biodegradable, non-toxic and biocompatible polymer [30]. This polymer has been demonstrated to have a distinct antimicrobial effect on the growth of mycelia, sporulation, spore viability and germination as well as the generation of fungal virulence factors in fungi [31]. Chitosan molecules can be used in agriculture for crop protection as well as plant disease control [32]. Although chitosan’s antimicrobial activities have been well researched, it is believed that it can serve as a chelating agent for the minerals and nutrients that pathogens require in order to survive [33]. Factors such as chitosan’s acetylation level, concentration, mode of application, target microbe and molecular weight are known to affect its antifungal efficacy [34]. Chitosan can be utilized as an efficient anchoring agent for metal nanoparticles due to the high concentration of -NH_2_ and -OH groups in its chemical composition (NPs). Antimicrobial activity of Cu NPs has been demonstrated against a wide range of microbiological species [35]. Because of their unusual property—a high surface area to volume ratio—Cu NPs have greater antimicrobial activity than copper salt [36]. However, its agglomeration in pure state pure may inhibit or decrease its antimicrobial activity. Recently, different reports suggested that incorporating Cu NPs into carrier agents has been shown to greatly increase its antimicrobial property [37,38,39,40]. In our research, we became interested to determine fungicidal effect of chitosan-decorated copper oxide nanoparticles, (CH@CuO NPs) against *B. cinerea*, the causal agent of tomato gray mold and sclerotia formation. To our understanding, antifungal action of pure copper-chitosan nanocomposite particularly its stability under high temperature degrees against *B. cinerea* infecting tomato is not previously studied.

Thus, the primary objectives of this research were to: (1) develop a pure nanocomposite from chitosan and copper oxide nanoparticles; (2) analyze the antifungal properties of CH@CuO NPs nanocomposite to control gray mold disease under laboratory and greenhouse conditions, (3) testing the stability of CH@CuO NPs antifungal activity under high temperature degrees, and (4) explore the possible mechanisms of CH@CuO NPs nanocomposite in controlling tomato gray mold.

## 2. Materials and Methods

### 2.1. Materials Used in This Study

Chitosan (CH) (molecular weight: 190–370 kDa, degree of de-acetylation: ≥75%), Potato dextrose agar (PDA), sodium tripoly phosphate (STPP), streptomycin sulfate, phosphate buffer solution (PBS), and propidium iodide were all purchased from Sigma Aldrich (Saint Louis, MO, USA). Tween 80 (Xilong Scientific Co., Ltd., Shantou, China), sodium hypochlorite (NaOCl) (Merck, Darmstadt, Germany), acetic acid (Merck, New York, NY, USA) and the fungicide Teldor 50% SC (Bayer Crop Science Co. (Cambridge, UK) were purchased from a commercial distributer. Double-distilled water was used throughout the entire work. 

### 2.2. Isolation and Molecular Characterization of B. cinerea 

Leaves of tomato plants exhibiting gray mold symptoms as pale brown to dark lesions were collected during the winter of 2021. Diseased leaves were thoroughly cleaned with sterile distilled water after being surface-disinfected in sodium hypochlorite (NaOCl, 2%) for 2 min and cut into small pieces. Isolation was carried out on potato dextrose agar (PDA) medium supplemented with streptomycin sulphate (300 mg/L). Fungal cultures were then purified and incubated at 16 °C in the dark for 4 days for mycelial growth. All the isolated fungi were initially identified using a morphological identification key [41,42]. 

Identity was further confirmed after PCR amplification and DNA sequencing of putative isolates of *B. cinerea*. DNA was extracted from approximately 50 mg of fresh mycelia of 7-days old cultures of *B. cinerea*. Extraction was performed using DNeasy plant mini kit (Qiagen, Hilden, Germany), following the manufacturer guidelines. PCR amplification and sequencing of the rDNA internal transcribed spacer (ITS) region was performed using the ITS1 and ITS4 primers [43]. PCR reactions and amplification conditions were performed as previously described [44]. The resulting PCR products were purified and sequenced at Macrogen (Gangnam-gu, Seoul, Republic of Korea South Korea In.). The generated sequences were then edited and corrected where necessary using MEGA 7 v. 11 [45], and homology was compared though a blast search tool with a database hosted by NCBI (http://www.ncbi.nlm.nih.gov/, accessed on 2 January 2023).

### 2.3. Preparation and Characterization of CH@CuO NPs

Chitosan nanoparticles (CH NPs) were prepared based on the ionic gelation of chitosan with sodium tri-polyphosphate (STPP) anions [46]. Chitosan was stirred in 1% (*v*/*v*) of acetic acid at 400 rpm in a magnetic stirrer overnight, and then filtered through a Polyvinyl Difluoride (PVDF) syringe filter with a 0.22 µm pore size. At the same time, STPP was prepared a concentration 0.25% (*w*/*v*) in a sterile distilled water and then filtered using a PVDF syringe filter drop by drop, with a magnetic stirrer rotating at 800 rpm, chitosan and STPP of equal volume were cross-linked, and followed by a 10-min centrifugation at 12,000 rpm. The obtained formulation was re-suspended in sterile distilled water and ultra-sonified at a 28% pulse ratio for 100 s at 4 °C to produce pellets. The CH NPs were purified and evenly dispersed through centrifugation and ultra-sonication, which was performed three times. Chemical precipitation was used to create copper oxide nanoparticles (CuO NPs), which was the second step in the synthesis process following a standard procedure described in previous work [25].

Preparation of CH@CuO NPs was carried out by mixing a 0.5 g of the prepared CuO NPs with 40 mL of the prepared CH NPs solution and flaked into a homogeneous suspension after sonication for 60 min. The obtained product was constantly washed with double-distilled water, centrifuged at 8000 rpm and dried in a vacuum oven. Subsequently, the product was stored into storage at 4 °C for future analysis. Characterization of the nanocomposite was carried out using transmission electron microscope (TEM) (TECNAI 10) (TEM, Philips, Amsterdam, The Netherlands). Samples were prepared by diluting one milligram of CH, CuO and CH@CuO NPs separately in one milliliter of distilled water; then, a single drop of the solution was sonicated for 1 h, then placed onto carbon coated copper TEM grids that had been previously coated and allowed to dry at room temperature for 3 h, while the extra solution was erased using a blotting paper. Fourier transform infrared (FTIR) spectrophotometry (Avatar-300, Nicolet, USA) analysis was conducted to determine the functional groups. The chemical functional groups responsible for the interaction of the CH NPs with the CuO NPs were detected at wavelength ranges from 4000–400 cm^−1^ with 16 average scans.

### 2.4. Effect of CH@CuO NPs on the Mycelial Growth of B. cinerea

The antifungal activity of CH@CuO NPs was evaluated against *B. cinerea* isolate using the poison food technique [47]. Briefly stated, three distinct concentrations of the synthesized CH@CuO NPs (50, 100 and 250 mg/L) were added to the PDA medium. Mycelial discs (5 mm) were picked using a sterilized cork borer and placed into the center of the PDA petri dishes containing CH@CuO NPs and incubated at 20 °C. Untreated and treated PDA plates with the commercial fungicide Teldor 50% SC at dose 1.5 mL/L were also used as negative and positive controls, respectively. Finally, the radial growth of fungal hyphae was measured in each inoculated plate to assess the antifungal effect of various concentrations of CH@CuO NPs using the following formula.
Inhibition (%) = [(T − t)/T] × 100
where T represents the mycelial growth of *B. cinerea* in the control plate and t represents the mycelial growth in the plate supplemented with CH@CuO NPs. Experiments were repeated in triplicate under sterilized conditions.

### 2.5. Effect of CH@CuO NPs on the Germination of Conidia and Sclerotia of B. cinerea

The inhibitory effect of CH@CuO NPs on the germination of *B. cinerea* conidia and its formation for germ tubes were investigated using the protocols outlined [48]. Firstly, *B. cinerea* was grown on PDA for 21 days at 20 °C. Conidia germination was verified if the germ tube length was equal to or greater than the conidia length [49]. PDB medium was supplemented with the concentrations 50, 100 and 250 mg/L of CH@CuO NPs. Conidial suspension was prepared by flooding the mycelial mat with sterile water amended with 0.5% Tween 80, and the appropriate conidia concentration was adjusted to 1 × 10^6^ conidia/mL using a hematocytometer. On a sterile glass slide, 15 µL of each CH@CuO NPs concentration were agitated with 3 µL of *B. cinerea* conidial suspension. Instead of CH@CuO NPs, sterilized PDB was used as a control. In order to maintain an elevated relative humidity, all slides were placed in Petri plates containing moist blotter paper. They were then incubated for 8 h at 20 °C under white radiant light. Germinated conidia were counted, and germ tube length was determined using a microscope (Olympus CX41, Japan). Following that, the germination inhibition percentage of conidia was calculated as described in Youssef et al. [26]. At least 50 fungal conidia were inspected for each treatment to assess the percentage of germination, and about 60 fungal conidia were used for germ tube dimension when more than 90% of *B. cinerea* conidia in the PDB control germinated after 8 h of incubation. This experiment was repeated three times for each treatment with three replicates. 

The effect of CH@CuO NPs on the sclerotia germination of *B. cinerea* was investigated using black developed sclerotia as described by Yang et al. [50]. The surface-sterilized sclerotia were treated with CH@CuO NPs by socking for 24 h at the tested concentrations, before being air-dried and placed in PDA plates. Each treatment was repeated three times, with 50 purposeful sclerotia in each. All PDA plates containing sclerotia were incubated for two days at 20 °C. Sclerotia germination rate was estimated after counting the germinated sclerotia in both the control and the treated plates.

### 2.6. Effect of CH@CuO NPs on the Plasma Membrane Integrity of B. cinerea Conidia

In this experiment, the conidia of *B. cinerea* (1 × 10^6^ conidia/mL) were treated with CH@CuO NPs at concentrations 50, 100 and 250 mg/L and incubated at 20 °C without agitation. The control treatment was replaced with sterilized distilled water instead of CH@CuO NPs. After that, the fungal conidia were centrifuged at 10,000 rpm after 2, 4, 6 and 8 h of incubation and stained and inspected using the method described in Wang et al. [5]. Briefly, the fungal conidia were stained for 5 min at room temperature with 10 g/mL of propidium iodide and then centrifuged. Then, the conidial pellets were washed with phosphate buffer solution (PBS) (pH 7.0) twice. Epifluorescence optics with blue excitation (450 to 490 nm) and 526 nm barrier filters were used for the inspections using a Nikon Eclipse 80i (Tokyo, Japan) microscope. The percentage of membrane veracity (MV) damage was calculated in triplicates for 200 of randomly selected fungal conidia using the following the formula:MV = (number of total examined fungal conidia − number of stained fungal conidia)/(number of total examined fungal conidia) × 100%).

### 2.7. Effect of CH@CuO NPs on the Mycelial Electrolyte Leakage of B. cinerea

Following cultivation of *B. cinerea* on PDA plates, three fungal mycelial discs (5 mm in diameter) were gently cut from the edges of the fungal colony using a sterile cork borer and transferred into potato dextrose broth (PDB) medium and incubated for 4 days at shaking rate of 200 r/min. Fungal mycelia from the culture medium were harvested, washed three times in sterile distilled water, filtered, and then weighted. Both the CH@CuO NPs and the chemical fungicide used as a positive control were diluted to a concentration of 50 µL/mL in sterile distilled water. The conductivities of the solutions were measured using a DDS-11C model conductivity detector at intervals ranging from 0 to 100 h after adding 1.0 g of fresh fungal mycelia to each of the solutions mentioned above. The experiment was repeated twice in triplicate per treatment. 

### 2.8. Effect of Heating on the Stability of CH@CuO NPs against B. cinerea

The nanocomposite CH@CuO NPs at 100 mg/L as a representative concentration were subjected to different temperatures 20, 40, 60, 80 and 100 °C for 15 min to examine their stability. After being heated, CH@CuO NPs were quickly cooled to room temperature and tested for antagonistic activity against *B. cinerea*. The antifungal effects of heat-treated CH@CuO NPs on the radial development of *B. cinerea* were investigated following the published procedure [51].

### 2.9. Effect of CH@CuO NPs on the Gray Mold Disease 

#### 2.9.1. Detached Tomato Leaves Experiment

Tomato leaves of five-week-old plants were detached and surface-disinfected by 1% NaOCl for 2 min. The leaves were then properly washed with sterile, distilled water and gently dried. Tomato leaves were then treated with 10 µL of CH@CuO NPs and left to dry in a laminar airflow. Control tomato leaves were sprayed with only distilled water. Tomato leaves were sprayed with 15 µL of *B. cinerea* conidia suspension (1 × 10^6^ conidia/mL) supplemented with 0.5% Tween 80 and 1 mg/mL of sucrose. The inoculated tomato leaves were incubated at 20 °C for 4 days, or until the gray mold lesions began to appear. The developed lesions were then measured and the efficacy was determined in triplicate as following: (lesion diameter in the control-lesion diameter in the treatment)/lesion diameter in the control × 100 [52].

#### 2.9.2. Ripe Tomato Fruit Experiment

Ripe tomatoes of uniform size and color, with no signs of rot, were employed. Inoculation procedure was carried out as described by Gao et al. [53]. Mature tomatoes were first soaked in a 1% (*w*/*v*) NaOCl solution for 2 min in order to decontaminate any microbial cells on their surfaces before being rinsed and allowed to air dry. Sterile forceps were used to make puncture wounds in tomato fruit (2 mm deep and 2 mm in diameter). The wounds were then treated with 10 µL of CH@CuO NPs, and control fruit were treated with sterile PDB. A 15 µL of the conidial suspension (1 × 10^6^ conidia/mL) was injected into each wound approximately 2 h after treatment with CH@CuO NPs. To provide a high relative humidity throughout the incubation process, each tomato fruit was placed on a moist absorbent paper in plastic boxes. After for 4 days of incubation, the lesion sizes were assessed [54]. There were three replicates for each treatment and each replicate contained three fruits. The experiment was repeated in triplicate.

### 2.10. Greenhouse Experiments

#### Effect of CH@CuO NPs on Tomato Gray Mold

Tomato seeds cv. Super strain B were seeded in vermiculite soil for four weeks. Seedlings were then transplanted in individual plastic pots containing mixed soil, irrigated and fertilized when needed. Two weeks later, tomato plants were sprayed with CH@CuO NPs at 50, 100 and 250 mg/L until they overflowed four hours before being inoculated with *B. cinerea* conidia. Tomato plants sprayed with PDB were served as a negative control. Four hours later, leaves were inoculated with 15 µL of conidial suspension (1 × 10^6^ conidia/mL) containing 1% sucrose solution and 0.1% Tween 80. The control plants were inoculated with sterile distilled water. The commercial fungicide Teldor (50% SC.) was employed as a positive control and applied with the recommended dose. The experimental plants were arranged out using a completely random block pattern. Three replicates were used for each treatment (3 plants for each). Four days after inoculation, gray mold symptoms were observed. 

### 2.11. Data Analysis

The obtained data were subjected to a one-way analysis of variance (ANOVA) [55]. Means were compared using the Least Significant Difference (LSD) test at (*p* < 0.05) using SPSS software v. 8.0. All experiments were repeated three times, and the data presented here are the average of the three sets (SD). GraphPad prism 9 software was utilized for creating and editing graphs. 

## 3. Results

### 3.1. Characterization of B. cinerea

The isolated *B. cinerea* were initially identified based on its morphological traits (Appendix A). On PDA plates, *B. cinerea* mycelia develop pale brown to grey conidia in mass as smooth. Conidia were unicellular, colorless, ellipsoidal to globose single cells that formed tree-like conidiophores with an average size of 9.7 × 9.3 µm. Conidiophores were brown, arose alone or in clusters, were straight or flexuous, septate, and ranged in length from 100 to 400 µm. After 4 weeks, black, spherical sclerotia (0.5 to 3.0 mm) were developed, immersed in the PDA, and distributed irregularly across the colonies. Blast search tool of the sequenced PCR product showed a 99.63% similarity with *B. cinerea* strain CBS 261.71. The generated sequence was deposited in NCBI database under accession number OQ134873.

### 3.2. Synthesis and Characterization of CH@CuO NPs

The resulting TEM images revealed that CH NPs have a thin and semitransparent network shape (Figure 1A), while CuO NPs were spherically shaped (Figure 1B). Furthermore, CH@CuO NPs exhibited an irregular shape (Figure 1C). Many dark spots were firmly immobilized on CH nanoparticles (Figure 1C). The size of the synthesized CH NPs, CuO NPs and CH@CuO NPs as measured through TEM, were approximately 18.36 ± 1.2 nm, 20.01 ± 2.4 nm, and 29.98 nm ± 1.4 nm, respectively. According to the FTIR results, the spectrum of CH contained several peaks representing functional groups, such as the stretching vibration peaks of O-H (3427 cm^−1^), C=C (1655 cm^−1^), and C-O (1286 cm^−1^), the asymmetric stretching vibration of C-H bonds in -CH3 and -CH2 (2924 cm^−1^) and the in-plane bending vibration of C-H (1425 cm^−1^) (Figure 1D).

### 3.3. Antifungal Activity of CH@CuO NPs against Mycelial Growth and Sclerotia Formation of B. cinerea

On potato dextrose agar (PDA), *B. cinerea* was cultivated with three different concentrations: 50, 100 and 250 mg/L of CH@CuO NPs. The findings demonstrated that, in comparison to fungal colonies produced on PDA plates that had not been treated (serving as control), all fungal colonies formed in PDA plates and altered with the three CH@CuO NPs concentrations were significantly smaller (Figure 2). Moreover, the inhibitory effect of CH@CuO NPs against *B. cinerea* significantly (*p* < 0.01) increased with an increase in concentration of CH@CuO NPs (Figure 2). As example, the results showed that CH@CuO NPs at a concentration of 50 mg/L showed 62% growth inhibition to *B. cinerea*, and an inhibition rate of 80% was recorded when treated with 100 mg/L, while the highest growth inhibition (92%) was recorded at 250 mg/L of CH@CuO NPs compared to the chemical fungicide treatment showing an inhibition rate (78%). This is in comparison to treating with naked chitosan (CH) and copper oxide (CuO) alone (with the same concentrations used in production CH@CuO NPs) showing only 29%, and 30% inhibition growth values respectively. 

The results also indicated that CH@CuO NPs had a remarkable inhibitory effect on sclerotia formation of the *B. cinerea* compared to the fungicide used (Figure 2). In this regard, CH@CuO NPs at conc. 250 mg/L showed the highest inhibition in sclerotia (98%) in relative to fungicide treatment (67%). At 21 dpi after treating *B. cinerea* with CH@CuO NPs at this concentration, a relatively lower sclerotium production was detected. This explains that CH@CuO NPs, at 250 mg/L conc., are able to reduce *B. cinerea*’s mycelium development and sclerotium formation more effectively than the commercial fungicide. In contrast to treating with CH & CuO alone showing little remarkable inhibitory effect with 20%, and 25% values respectively on sclerotia formation of the *B. cinerea*. 

### 3.4. CH@CuO NPs Inhibited Conidia Germination of B. cinerea

The results showed that the three tested concentrations of CH@CuO NPs significantly inhibited conidia germination and sclerotia formation, and both of them positively interacted with the CH@CuO NPs concentration (Figure 3A). Specifically, at 8 h post-inoculation (hpi), the germination percentage of conidia was above 95% in the control, while only about 20% germinated in CH@CuO NPs at 50 mg/L, and the germ tube length (average 5.1 µm) of the germinated conidia was much shorter than that of the control *B. cinerea* (average 25.8 µm). Notably, the lowest geminated conidia were detected at concentrations 100 and 250 mg/L of CH@CuO NPs giving 85%, 95% inhibition respectively at 8 hpi (Figure 3B). Moreover, number of sclerotia formed was recorded in treatments with 250 mg/L CH@CuO NPs (Figure 3A). 

### 3.5. Effect of CH@CuO NPs on B. cinerea’s Mycelial Electrolyte Leakage

To examine the effect of CH@CuO NPs at different concentrations on the fungal cell membrane, the electrical conductivity of mycelia suspension was measured (Figure 4). The conductivity of fungal mycelia suspension treated with CH@CuO NPs with varied values and the chemical fungicide greatly increased compared with the conductivity of the control mycelia at all treatment times. CH@CuO NPs induced more significant electrolyte leakage from hyphae than Teldor 50% SC fungicide.

### 3.6. Antifungal Activity of CH@CuO NPs Retains Its Activity at High Temperatures

CH@CuO NPs at 100 mg/L was selected as an example to examine the potential antifungal activity of the produced CH@CuO NPs at high temperatures. The results demonstrated that both heat-treated CH@CuO NPs and non-heat-treated CH@CuO NPs cultural plates inhibited *B. cinerea* mycelial growth, with no significant differences identified between the treatments. In this regard, the highest level of fungal growth inhibition rate (%) against the tested *B. cinerea* isolate was obtained with significant values of 90.42 ± 1.25, 91.22 ± 2.1, 90.80 ± 0.33 and 90.8 ± 2.5%, for CH@CuO NPs heated to 40, 60, 80 and 100 °C respectively, compared to non- heated CH@CuO NPs treatment which showed approximately the same value (91.80 ± 2.1), after eight days of incubation (Figure 5). 

### 3.7. Fungicidal Activity of CH@CuO NPs in Reducing/Controlling Gray Mold Disease on Detached Tomato Leaves and Their Fruits 

The response of artificially inoculated tomato leaves with *B. cinerea* after treatment with three different concentrations of CH@CuO NPs as well as the chemical fungicide treatment are expressed in Figure 6. Typical lesions of *B. cinerea* were observed on tomato leaves in some treatments. The lesion area developed by *B. cinerea* measured about 2.5 cm at 4 dpi, but this area was significantly reduced to 0.4, 0.1 and 0 cm in diameter when leaves were sprayed with CH@CuO NPs at concentrations 50, 100 and 250 mg/L, respectively compared to the treatment with the control (Figure 6A,B). Disease symptoms were also reduced when tomato plants were treated with a conventional fungicide (Teldor 50% SC), but not as effectively as “a lesser extent” when treated with CH@CuO NPs, particularly at concentrations 100 and 250 mg/L.

Furthermore, our study also investigated the potential activity of CH@CuO NPs to control gray mold on tomato fruits (Figure 6C,D). Throughout this experiment, we observed no *B. cinerea* lesions on the surface of inoculated tomato fruits sprayed with CH@CuO NPs at the tested concentrations (Figure 6C,D), whereas 4 days after inoculation with *B. cinerea* conidia, the lesion diameter on tomato fruits treated with 50 mg/L of CH@CuO NPs was 0.7 cm, which was only slightly smaller than the lesion diameter on *B. cinerea* treated fruits (2.9 cm) (Figure 6C,D). However, no lesions were observed on tomato fruits treated with CH@CuO NPs at con. 100 and 250 mg/L compared to 0.5 com lesions in case of fruits treated with the chemical fungicide. This suggests that CH@CuO NPs significantly controlled/reduced the lesions caused by *B. cinerea* on both leaves and fruits, particularly at concentrations 100 and 250 mg/L of CH@CuO NPs. Furthermore, the potential of CH@CuO NPs to control tomato gray mold was evaluated on whole tomato plants (Figure 7A,B). The efficiency was shown to be 100% when plants were treated with both 100 and 250 mg/L of CH@CuO NPs against *B. cinerea* when compared with the untreated control (Figure 7). 

## 4. Discussion

Gray mold disease caused by *B. cinerea* is the most significant foliar and fruit disease affecting tomatoes. Fungicide application is the primary method for controlling this disease. However, the development of single-site inhibitor resistance by this pathogen needs research into the creation of unique, biodegradable, efficient and safe products. In this study, we investigated the antifungal properties of CH@CuO NPs against the causal agent of tomato gray mold *B. cinerea* throughout in vitro and in vivo trials. The identity of isolated *B. cinerea* was confirmed though BLASTn analysis of the sequence against reference sequences in GeneBank database. The obtained sequence exhibited 99.63% similarity with GeneBank *B. cinerea* strain CBS 261.71. The partial sequences of the ITS regions from *B. cinerea* isolate was deposited in GeneBank under accession No. OQ134873.1.

Based on the obtained results of TEM analysis, the size of the formed CH, CuO and CH@CuO NPs reached 18.28± 2.4 nm, 19.34 ± 2.1 nm and 32.74 ± 2.3 nm, respectively. Different reports indicated that such small particles may enhance their antimicrobial properties due to more interaction chances with microbial cells [56,57]. According to FT-IR results, the primary functional group of chitosan showed a peak at 3427 cm^−1^, which is attributed to the O-H group of stretching vibrations. The N-H bending vibration of the protonated amino (-NH_2_) group and the C-H bending vibration of the alkyl group are responsible for the tiny absorption peaks at 1630 cm^−1^. Due to the anti-symmetric stretching vibration of C-O-C bridges, the tiny absorption peaks at 1059 cm^−1^ are identified and attributed to the glucopyranose ring in the chitosan matrix. CuO NPs’ spectral peaks shared a similar sharpness and a relative shift at 1631 and 1536 cm^−1^. However, the current findings support a little fluctuation, suggesting that Cu bonding affects the variation. After the reaction between CH and CuO NPs, a noticeable reduction in the percentage of chitosan NPs transmission was observed, where the strong peak at 3427 cm^−1^ was significantly reduced and shifted into O–H stretching vibration at 3446 cm^−1^ and 3354 cm^−1^ [58,59]. The adsorption band of C=C at 1655 cm^−1^ drifted to 1624 cm^−1^ [60]. The C–H out-planes bending vibrations at 987 stretching vibration in monoclinic CuO below 600 cm-1 [61]. These noticeable modifications to the functional groups indicated that CuO NPs had successfully immobilized on the surface of CH NPs.

On the other hand, the inhibitory action of CH@CuO NPs against *B. cinerea* significantly showed a dose-dependent decrease in mycelial growth. For example, the highest growth inhibition (92%) was recorded at 250 mg/L, followed by 100 mg/L which exhibited an inhibition rate of 80%. By contrast, the chemical fungicide revealed an inhibition rate of 78%, while the lowest concentration 50 mg/L of CH@CuO NPs showed 62% inhibition to *B. cinerea*. Moreover, CH@CuO NPs had an inhibitory effect on sclerotia formation of the *B. cinerea* at varied values, where at conc. 250 g/L CH@CuO NPs showed the highest inhibition in sclerotia (98%) in comparison to the chemical fungicide treatment (67%). This finding could possibly be explained by the smaller NPs’ increased surface area when compared to their bigger counterparts, which might encourage more interactions with the fungal cells and a greater and faster release of the copper ions [62,63]. Gray mold is thought to be spread mostly through fungal conidia. Therefore, evaluating whether or not *B. cinerea* conidial germination is inhibited by CH@CuO NPs provides a direct way to see if the plant disease may spread and become established. So, we tested the inhibitory activity of CH@CuO NPs at the three concentrations 50, 100 and 250 mg/L on the germination rate of *B. cinerea* conidia and to sclerotia formation. The results showed that the three tested concentrations of CH@CuO NPs significantly inhibited conidia germination and sclerotia formation, and both of them positively interacted with the CH@CuO NPs concentration. However, the lowest geminated conidia were detected at concentrations 100 and 250 mg/L exhibiting inhibition rates of 85% and 95%, respectively. Moreover, no sclerotia formation was recorded in treatments with 250 mg/L CH@CuO NPs. This indicates that CH@CuO NPs is crucial in preventing conidia germination and affecting the length of *B. cinerea*’s germ tubes. These findings are in line with those of other studies showing that nanocarrier systems and CuO NPs both exhibit multiple modes of inhibitory action to microbial pathogens [63], attempting to make them a promising alternative to traditional synthetic pesticides for the management of a wide variety of fungal pathogens that cause plant diseases. This would greatly aid in decreasing the need for hazardous pesticides, which are incredibly harmful around food crops such as fruits and vegetables [64,65]. The data obtained so far suggest that CH@CuO NPs are a suitable replacement for traditional fungicides in crop protection.

Furthermore, it was noticed that CH@CuO NPs induced more significant electrolyte leakage from hyphae than Teldor 50% SC fungicide. As a result, it was hypothesized that CH@CuO NPs contributed to membrane damage in cells. The sensitivity of hyphae to chemical substances including chemical fungicides is measured by their ability to cause electrolyte leakage. The CH@CuO NPs considerably increased the change in conductivity compared to the chemical fungicide, which has been thought to cause some membrane weaknesses but had no impact on ion leakage or water permeability [66]. The results revealed that CH@CuO NPs enhanced the conductivity of the solution, caused electrolyte leakage from the cell, and inflicted some damage on the mycelia cell membrane system. Thus, the biochemical alterations in the plasma lemma brought on by CH@CuO NPs were most likely to be linked to the breakdown of cell walls. We also suggested that CH@CuO NPs could reduce DNA and polysaccharide levels in *B. cinerea*’s intact mycelia [67].

In order to examine the potential antifungal activity of the produced CH@CuO NPs at high temperatures, CH@CuO NPs at 100 mg/L was selected in this regard. The results demonstrated that both heat-treated CH@CuO NPs and non-heat-treated CH@CuO NPs cultural plates inhibited *B. cinerea* mycelial growth, with no significant differences identified between the treatments. This indicates that the antifungal activity of CH@CuO NPs is able to retain its activity at high temperatures and suggests that the high temperature is not passively affecting on the polymeric nature of chitosan as a natural component. This unique property ensures its stability at high temperature when decorated with CuO nanoparticles, which can be exploited further in long term preservation of CH@CuO NPs as commercial fungicide under un-favorable storing conditions.

Furthermore, spraying the detached tomato leaves with CH@CuO NPs indicated that lesions of *B. cinerea* on those leaves were significantly reduced to 0.4, 0.1 and 0 cm in diameter compared to 2.5 cm in control tomato leaves at 4 dpi. Moreover, CH@CuO NPs significantly reduced the lesions on both leaves and fruits particularly at conc. 100 and 250 mg/L. Furthermore, the potentiality of CH@CuO NPs to control tomato gray mold was evaluated on whole tomato plants. This result is consistent with earlier studies, which indicated that CuO NPs were more effective than the other six metallic oxide nanoparticles (ZnO, TiO, AlO, FeO, NiO or MnO) at promoting tomato and eggplant growth in soil infected with *Verticillium dahliae* and *F. oxysporum* f.sp. *lycopersici*, respectively [68,69]. Furthermore, various grape components of the Sousao grape varieties treated with CH demonstrated higher antibacterial activity [70]. This issue also included field research that revealed CH may be utilized to create new plant protection solutions, either independently or in combination with other ingredients, to control diseases in a sustainable manner. For instance, copper-based compounds are frequently employed in grapevine-control strategies, but the application is constrained by European law, necessitating the urgent need to develop a different approach [70].

By analyzing the data collected, we hypothesize that the fact that tomato grey mold infection occurs on young seedling plants early in the season may be one explanation for the long-lasting advantages of CH@CuO NPs [71], emphasizing the significance of a disease/treatment window. Host defense may considerably prevent or inhibit grey mold infection if plant cells have enough CH@CuO NPs available, as well as delay the emergence of grey mold disease symptoms to the point that the disease does not significantly spread. Another potential mechanism is the up-regulation of host defense-related genes by CH@CuO NPs. Tomato plants would be better able to handle *B. cinerea* infection as a result. Our findings are encouraging, but they do not definitively explain the precise mechanism by which CH@CuO NPs control the fungus. Elmer et al. [69] hypothesized that CH@CuO NPs can translocate into other areas of the treated plants and pass through root cells, causing systemic resistance.

## 5. Conclusions

Application of nanotechnology in agriculture could provide a safe and effective way to combat plant diseases. In this study, we tested different concentrations of a chitosan- CH@CuO NPs nanocomposite for their antifungal potential against *B. cinerea* the causal agent of tomato gray mold. Initially, spectroscopic and microscopic investigations were used to synthesize and characterize CH@CuO NPs. Validity of the nanocomposite’s antifungal activity was determined through in vitro and in vivo trials. At incredibly low concentrations, the CH@CuO NPs have an excellent antifungal activity. The lethal effect of CH@CuO NPs on fungal spore’s germination and sclerotia formation was confirmed by fluorescence analysis. Importantly, in vitro and in vivo assays on either detached tomato leaves, whole plants or their fruits artificially infected by gray mold disease demonstrated that CH@CuO NPs improved a powerful antifungal efficacy at concentrations of 100 and 250 mg/L. A standard chemical fungicide was utilized for comparison purposes. Finally, because of their biodegradability and biocompatibility, CH@CuO NPs may show to be a useful biomaterial for creating unique crop protection strategies for vintners all over the world, as well as potentially being utilized as safer bio-pesticides in place of synthetic pesticides. Despite this, the precise molecular mechanism underlying the interaction between chitosan and the host plant is still unknown. Nevertheless, a thorough transcriptome, metabolomics, and proteomic approach may reveal the fundamental molecular mechanism, enabling the development of its application strategies. Because of this, much more information will eventually be needed to fully understand how this nanocomposite interacts in regard to the structure-activity relationship, scale up the synthesis procedure, and produce stable formulations that have the desirable characteristics.

## Figures and Tables

**Figure 1 polymers-15-01099-f001:**
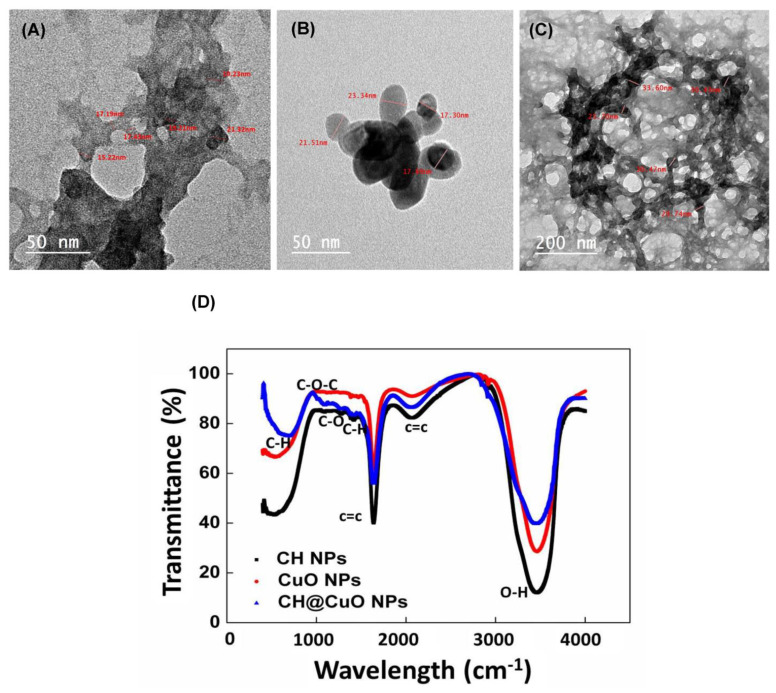
Transmission electron microscope (TEM) images of CH NPs (**A**), CuO NPs (**B**), CH@CuO NPs (**C**) and the FTIR spectrum of CH contained several peaks representing functional groups (**D**).

**Figure 2 polymers-15-01099-f002:**
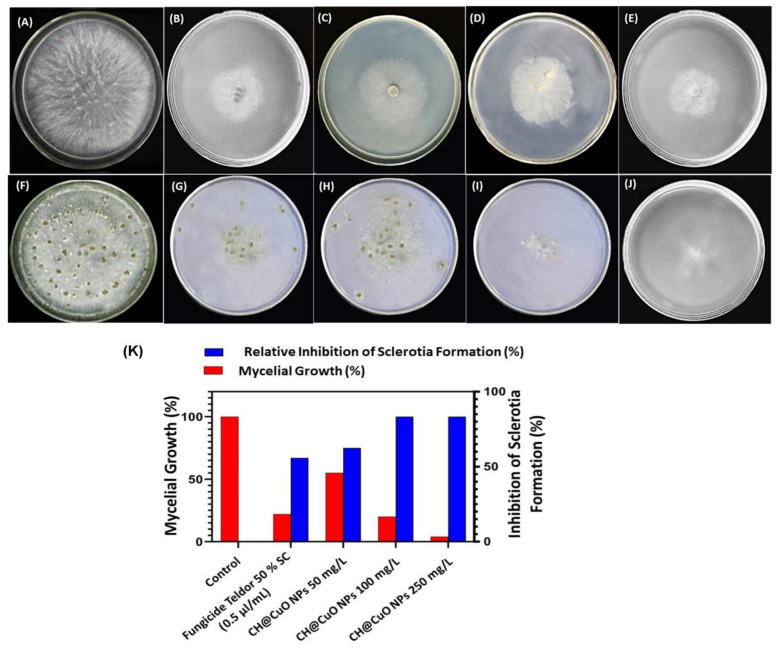
Inhibitory activity of CH@CuO against *B. cinerea* mycelial growth and sclerotia formation on PDA medium; non treated plates cultivated with *B. cinerea* (control) (**A**); plate treated with Teldor 50% SC (**B**); treated plates with CH@CuO at 50 (**C**),100 (**D**) and 250 mg/L (**E**) after 8 days of incubation at 20 °C; no treated plates cultivated with *B. cinerea* (control) (**F**); *B. cinerea* plate treated with Teldor 50% SC (**G**); plates treated with CH@CuO at 50 (**H**), 100 (**I**) and 250 mg/L (**J**) after 21 days of incubation at 20 °C. Mycelial growth of *B. cinerea* and sclerotia (**K**) in response to treatments with CH@CuO NPs and the fungicide (Teldor 50% SC.).

**Figure 3 polymers-15-01099-f003:**
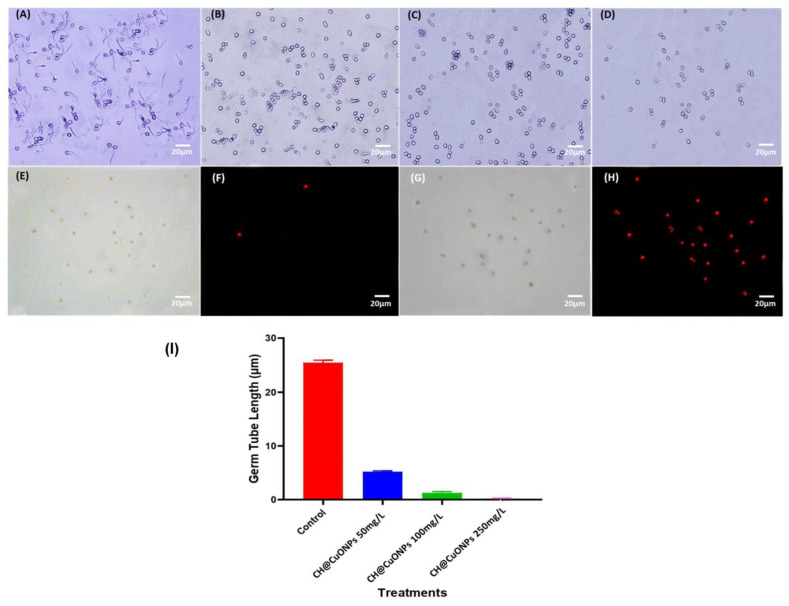
Effect of CH@CuO NPs concentrations on *B. cinerea* conidial cells treated with sterile water (control) (**A**), (**B**–**D**) incubated in CH@CuO NPs at 50, 100 and 250 mg/L for 8 h, respectively; (**E**) conidial cells of *B. cinerea* incubated in sterile water for 8 h and stained by propidium iodide (control) (**F**–**H**) conidia cells of *B. cinerea* stained by propidium iodide after treatment for 8 h with CH@CuO NPs at 50, 100 and 250 mg/L, respectively. (**I**) Germ tube length of *B. cinerea* in response to treatment with CH@CuO NPs at concentrations 50, 100 and 250 mg/L compared to control.

**Figure 4 polymers-15-01099-f004:**
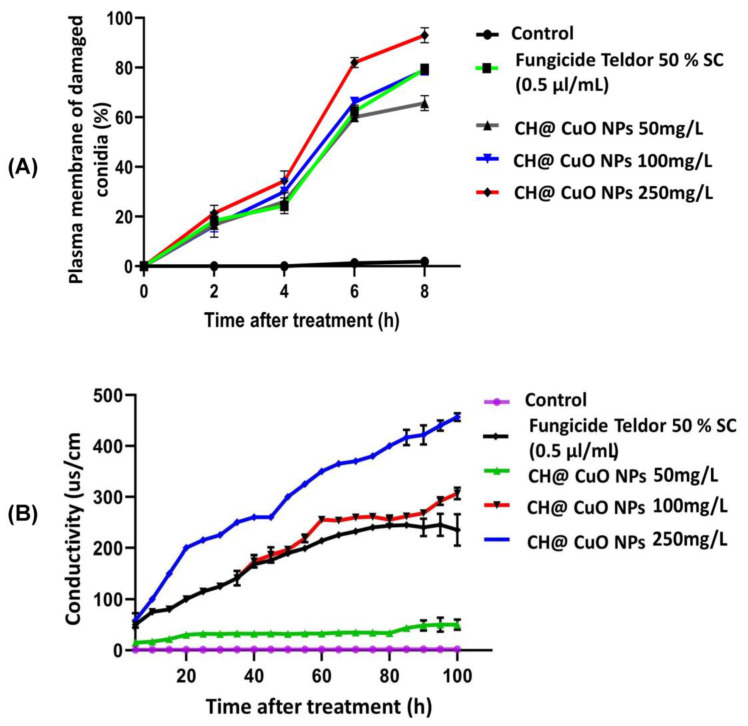
(**A**) The percentage of plasma membrane damaged conidia after treatment with CH@CuO NPs at different concentrations 50, 100 and 250 mg/L in comparison to Teldor 50% SC. (**B**) Electrolyte leakage from *B. cinerea* suspension during different time exposure to different concentrations of CH@CuO NPs compared to the fungicide.

**Figure 5 polymers-15-01099-f005:**
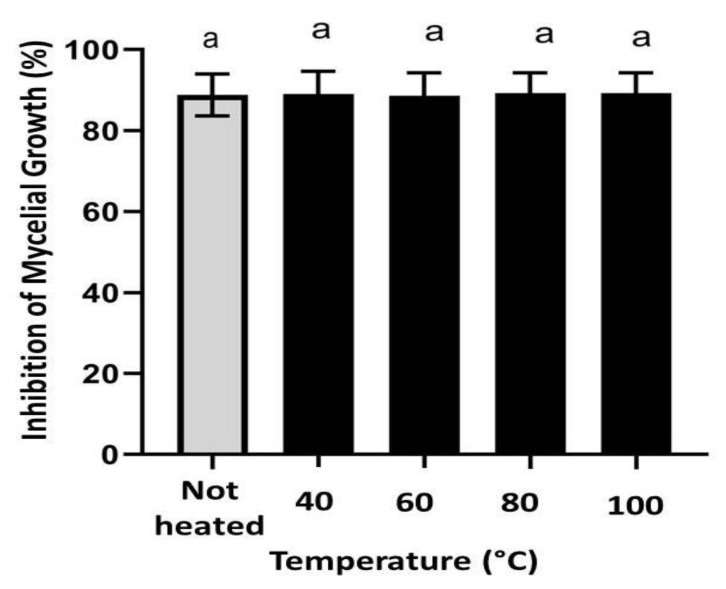
Effect of heating on the antifungal activity of CH@CuO NPs at 100 mg/L as representative conc.) against *B. cinerea* growth. Bars with the same letters are not statistically significant based on the least significant difference (LSD) test (*p* < 0.01).

**Figure 6 polymers-15-01099-f006:**
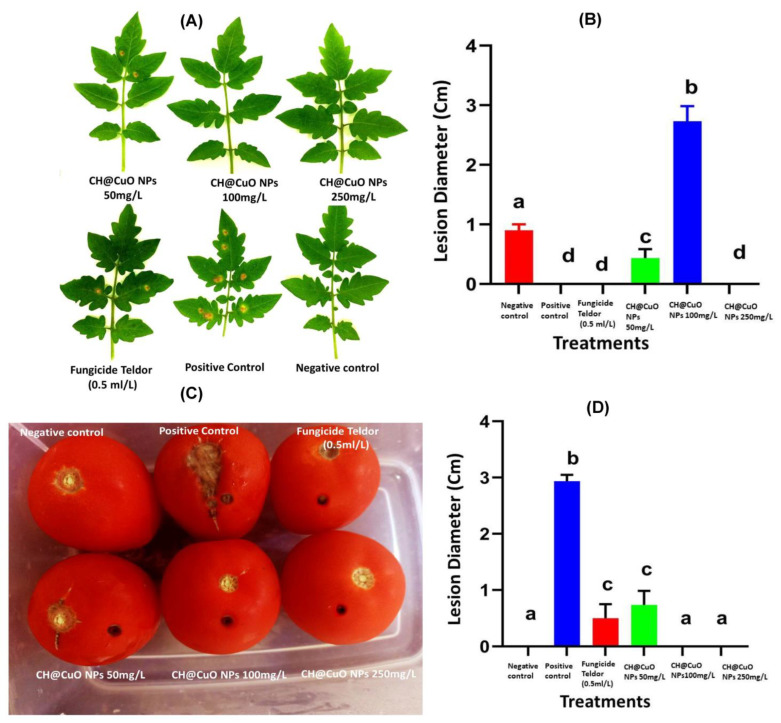
Effect of CH@CuO NPs on gray mold on tomato detached leaves treated with 50, 100 and 250 mg/L of CH@CuO NPs and fungicide (Teldor 50% SC.) (**A**,**B**); Effect of CH@CuO NPs on gray mold on tomato fruits, treated with 50, 100 and 250 mg/L of CH@CuO NPs and fungicide (Teldor 50% SC.) (**C**,**D**). Bars with the same letters are not statistically significant based on the LSD test (*p* < 0.01).

**Figure 7 polymers-15-01099-f007:**
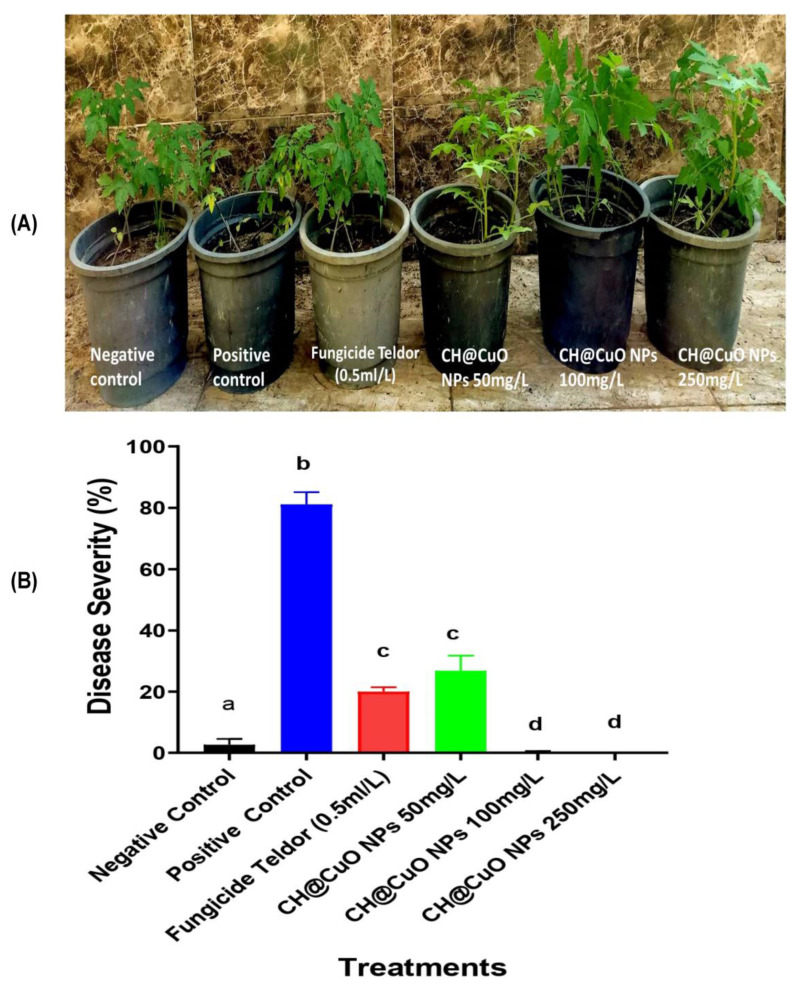
Gray mold disease severity on leaves of whole tomato plants in response to treatment with CH@CuO NPs at concentrations 50, 100 and 250 mg/L and Teldor 50% SC (**A**,**B**). Bars with the same letters are not statistically significant based on the LSD test (*p* < 0.05).

## Data Availability

Not applicable.

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
