# Peer review of "Chitosan-Decorated Copper Oxide Nanocomposite: Investigation of Its Antifungal Activity against Tomato Gray Mold Caused by Botrytis cinerea"

_polymers, 2023, doi:10.3390/polym15051099_

Round 1

Reviewer 1 Report

In this paper authors proposed the preparation of chitosan decorated copper oxide nanocomposite (CH@CuO NPs) and investigated its antifungal properties to control the gray mold diseases of tomato caused by the Botrytis cinerea. As discussed, the treatment of CH@CuO NPs with certain concentrations could exhibit better control efficacy than the conventional chemical fungicide. From my point of view, the topic is interesting and meaningful in the field of sustainable agriculture, but the manuscript needs to be strongly improved with major revisions to be suitable for publication in Polymers.

1.    As stated in line 90-93, the study is aimed to address the aggregation of Cu NPs. However, as shown in Fig. 1, the CH NPs and CH@CuO NPs are not single nanoparticles. Does the load of Cu NPs on the CH nanocarriers improve the antifungal activity? The authors should provide related experiment and data.

2.    Line 100: This description is inaccuracy. The antimicrobial activity of chitosan and related formulations have been widely investigated.

3.    There are many mistakes in the manuscript. Please check and correct them. e.g., line 105, NH2-, OH-; line 175 cm1; line 187; line 249, gm/L; line 308,100 to 400 m; line 322, cm-1

4.    Line221:"50,100 and 250 mg/L" Please add a space between 50 and 100.

5.    Section2.9: Why do the authors study the effects of heat treatment on the stability of CH@CuO NPs? How do you choose these experiment temperatures?

6.    Please keep the unit unified. e.g., hours or h? minutes or min? ul or uL?

7.    Figures: Please check the marks in figure 2, figure 3, figure 6. e.g., 50 nm,100nm, 250 nm?

8.    Line33-534: The description is not in accordance with the picture (figure C and D). Please check and revise the discussion in line 532-549 accordingly.

9.    Line 716: B. cinerea should be B. cinerea.

Author Response

Dear Professor,

We greatly appreciate all the critiques and comments from you. Those comments are extremely helpful for us to improving our paper, and they provide valuable guidance for our future study. According to these comments, we have carefully improved our manuscript, and all the revisions are highlighted in yellow in the text. Please see below point-by-point responses to the comments:

Comments and Suggestions for Authors

In this paper authors proposed the preparation of chitosan decorated copper oxide nanocomposite (CH@CuO NPs) and investigated its antifungal properties to control the gray mold diseases of tomato caused by the Botrytis cinerea. As discussed, the treatment of CH@CuO NPs with certain concentrations could exhibit better control efficacy than the conventional chemical fungicide. From my point of view, the topic is interesting and meaningful in the field of sustainable agriculture, but the manuscript needs to be strongly improved with major revisions to be suitable for publication in Polymers.

Question 1:    As stated in line 90-93, the study is aimed to address the aggregation of Cu NPs. However, as shown in Fig. 1, the CH NPs and CH@CuO NPs are not single nanoparticles. Does the load of Cu NPs on the CH nanocarriers improve the antifungal activity? The authors should provide related experiment and data.

Response: Thank you for your valuable comments and suggestions. Firstly, Line 90-93 illustrate the susceptible of pure Cu NPs for aggregation in some preparation protocol. In Fig. 1B we produced copper oxide nanoparticles which is not agreegated however dome particles appeared in the image over each other in a horizontal  view. Then, CH@CuO NPs appearance in Fig. 1C is not an aggregation but it appears entrapping CuO NPs on the surface of Chitosan NPs network as a result of electrostatic attraction.  Secondly, we thank the reviewer for his accurate revision as we regret that we didn’t include more detailed description on the antifungal activity of both chitosan and copper oxide nanoparticles alone in the first  version manuscript. However, we’ve now provided more relevant results in the updated version showing the antifungal activity of both chitosan and copper oxide nanoparticles alone in compared to the final form of Ch@CuO NPs. (Highlighted: See line 374-377 & 385-386)

Question 2:  Line 100: This description is inaccuracy. The antimicrobial activity of chitosan and related formulations have been widely investigated.

Response: Thanks for your valuable comment. We have corrected the sentence.

Question 3:    There are many mistakes in the manuscript. Please check and correct them. e.g., line 105, NH2-, OH-; line 175 cm1; line 187; line 249, gm/L; line 308,100 to 400 m; line 322, cm-1.

Response: Thank you for picking up the typing  errors. We have carefully rechecked and corrected all of them through the text.

Question 4:  Line221:"50,100 and 250 mg/L" Please add a space between 50 and 100.

Response: We have checked and corrected.

Question 5:  Section2.9: Why do the authors study the effects of heat treatment on the stability of CH@CuO NPs? How do you choose these experiment temperatures?

Response: Many thanks for your kind comment. In this experiment, we just intend to highlight any changing in the antifungal activity of our chitosan based compound could be modified with temperature. This is because chitosan at high temperature in air undergoes degradation. Where, thermal analysis with a derivatograph showed that chitosan as a polymer cannot withstand high temperatures "as example it degrade at 200 ºC". (Diab et al. 2011).  Hence its biological activity many be changed at high temerpatures. For this reason we proposed a course of different temperatures "40-100" higher than normal room temperature to score any change in the antifungal activity at any of them. On the other hand, it was commonly known that chitosan with high molecular weight (MW) is regarded as more stable and vise versa (SzymaÅ„ska and  Winnicka, 2015). Moreover,  the MW was found to affect the thermal stability of the polymer. Taken all together, we feel we should  investigate the activity of chitosan as we are working on a low molecular weight chitosan.

Question 6: Please keep the unit unified. e.g., hours or h? minutes or min? ul or uL?

Response: Thanks for your kind comment. We have carefully rechecked and corrected the both the typing errors.

Question 7:   Figures: Please check the marks in figure 2, figure 3, figure 6. e.g., 50 nm,100nm, 250 nm?

Response: Thanks for your kind comment, we have rechecked.

Question 8:   Line33-534: The description is not in accordance with the picture (figure C and D). Please check and revise the discussion in line 532-549 accordingly.

Response: Thanks for your kind comment. We have carefully rechecked and corrected.

Question 9:    Line 716: B. cinerea should be B. cinerea.

Response: Thanks for your valuable comment and accurate revision. It was corrected.

Reviewer 2 Report

Authors investigated the antifungal properties of chitosan decorated copper oxide nanocomposite (CH@CuO NPs), providing a safe and effective way to combat gray mold diseases of tomato. The results are interesting and constructive for controlling plant fungal diseases. Some questions need to be answered.

1) The novelty of this work should be added in the Introduction. For example, what are the advantages of the designed CH@CuO NPs in this paper compared to other  CuO-based nano-compounds against major plant pathogenic fungi? The comparison and the advantages can be added to the manuscript.

2) How to prepare the TEM samples? Figure 1C, how to distinguish TPP-CH and CuO NPs? Details should be described.

3) the FTIR spectrum, whether CuO has characteristic peaks at ~500 nm? Large Line width may cover up the details.

4) Please provide attachment diagram for Characterization of B. cinerea

5) CuO is classical fungicides in plant disease control and CH also exhibited antifungal effect. Then, whether antifungal effect of CH@CuO NPs has synergistic effect? It is recommended to conduct CH and CuO as negative controls.

Author Response

Dear Professor,

We greatly appreciate all the critiques and comments from you. Those comments are extremely helpful for us to improving our paper, and they provide valuable guidance for our future study. According to these comments, we have carefully improved our manuscript, and all the revisions are highlighted in red in the text. Please see below point-by-point responses to the comments:

Comments and Suggestions for Authors

Authors investigated the antifungal properties of chitosan decorated copper oxide nanocomposite (CH@CuO NPs), providing a safe and effective way to combat gray mold diseases of tomato. The results are interesting and constructive for controlling plant fungal diseases. Some questions need to be answered.

Question 1: The novelty of this work should be added in the Introduction. For example, what are the advantages of the designed CH@CuO NPs in this paper compared to other  CuO-based nano-compounds against major plant pathogenic fungi? The comparison and the advantages can be added to the manuscript.

Thank you for your valuable comments, we have now provided the introduction section with some important sentences in order to more clarify the novelty of the manuscript. (See line 109-121).

Response:

Question 2: How to prepare the TEM samples? Figure 1C, how to distinguish TPP-CH and CuO NPs? Details should be described.

Response: Many thanks for your kind comments, we have now add some words descripting preparing TEM samples (see line 175-179). Also , in Figure IC, CuO NPs is presented by the dark spots on the Chitosan NPs irregular layers as it was dopped on its surface.

Question 3: the FTIR spectrum, whether CuO has characteristic peaks at ~500 nm? Large Line width may cover up the details.

Response: Many thanks for your advise. We reduced the size width in the graph according to you kind recommendation. Also, there is no strong absorption peak at 500 nm so we did not include it.

Question 4: Please provide attachment diagram for Characterization of B. cinerea.

Response: Thank you for your valuable comments. We have provided the manuscript with a diagrammatic image characterize B. cinerea infection life cycle and also its treatment using CH@CuO NPs.

Question 5: CuO is classical fungicides in plant disease control and CH also exhibited antifungal effect. Then, whether antifungal effect of CH@CuO NPs has synergistic effect? It is recommended to conduct CH and CuO as negative controls.

Response: Thank you for your valuable comments, we agree with you that CuO fungicides are belongs to classical fungicides. Our presented data indicates the appearance of a synergistic activity appeared when CuO decorated on CH nanoparticles and it was significantly improved against B. cinerea at laboratory. Following your kind recommendations, we’ve now provided more relevant results in the updated version showing the antifungal activity of both chitosan and copper oxide nanoparticles alone in compared to the final form of Ch@CuO NPs in vitro experiments (Highlighted: See line 381-384 & 392-393)

Reviewer 3 Report

In this work, the potential antifungal properties of decorated copper oxide nanocomposite to control gay mold disease ot tomato caused by Botrytis cinerea throughout in vitro and in vivo trials was investigated.

My observations are de the folowing:

1. It is recommended a genral revision; as an example, in the las paragraph of the discussion, change "we hypothsis" by we hypothesize"

2. I think, 75% is the deacetylation degree, instead of acetylation degree.

3. Provide the particle size of pristine CH

4. Provide the deacetylation degree as well as the moelcular weight of the CH NPs.

5. In figure 1C, the dark region corresponds to CH which has encapsulated a Cu NP (briliant); if this is correct, the Ch CuO NP size is not 32.74 nm

6. Is is recomended to diminish the line width in figure 1D; then discuss accordingly.

7. Have the authors an idea of the cost for commercial production of the proposed fungicide?

Author Response

Dear Professor,

We greatly appreciate all the critiques and comments from you. Those comments are extremely helpful for us to improving our paper, and they provide valuable guidance for our future study. According to these comments, we have carefully improved our manuscript, and all the revisions are highlighted in red in the text. Please see below point-by-point responses to the comments:

Comments and Suggestions for Authors

In this work, the potential antifungal properties of decorated copper oxide nanocomposite to control gay mold disease ot tomato caused by Botrytis cinerea throughout in vitro and in vivo trials was investigated.

Question 1: It is recommended a general revision; as an example, in the last paragraph of the discussion, change "we hypothsis" by we hypothesize".

Response: Thanks for your kind comment, we have rechecked and corrected. (See line 737)

Question 2: I think, 75% is the deacetylation degree, instead of acetylation degree.

Response: We agree with you in this. We also provided this information in Materials section (See line 120)

Question 3: Provide the particle size of pristine CH

Response: Thanks for your kind comment . We corrected the particle size of  chitosan nanoparticles in line 320 which is "18.36 ± 12.4 nm"

Question 4: Provide the deacetylation degree as well as the moelcular weight of the CH NPs.

Response: Many thanks for your comment. We have provided the deacetylation degree as well as the moelcular weight of the chitosan molecules in materials section (See Line 120).

Question 5: In figure 1C, the dark region corresponds to CH which has encapsulated a Cu NP (briliant); if this is correct, the Ch CuO NP size is not 32.74 nm.

Response: Thank you for your valuable comments , we corrected the average size into 29.98 nm. See line 320

Question 6: Is is recomended to diminish the line width in figure 1D; then discuss accordingly.

Response: Many thanks for your advise. We reduced the size width in the graph according to you kind recommendation.

Question 7: Have the authors an idea of the cost for commercial production of the proposed fungicide?

Response: In fact, we roughly calculated the commercial cost for large production of CH@CuO which is significantly cheaper than Teldor 50% SC by more than 10 times and many other recommended fungicides in the market.

Round 2

Reviewer 1 Report

Please check the marks again in figure 2, figure 3, figure 6. I think  50 nm,100 nm, and 250 nm should be  50 mg/L,100 mg/L, and 250 mg/L.  The authors should check the whole text again carefully according to the comments from the reviewers.

Author Response

Dear Professor,

We greatly appreciate all the critiques and comments from you. Those comments are extremely helpful for us to improving our paper, and they provide valuable guidance for our future study. According to these comments, we have carefully improved our manuscript. Please see below point-by-point responses to the comments:
